# Scalable Infomin Learning

**Yanzhi Chen**[1], **Weihao Sun**[2], **Yingzhen Li**[3], **Adrian Weller**[1,4]

[1]University of Cambridge, [2]Rutgers University, [3]Imperial College London, [4]Alan Turing Institute

## Abstract

The task of infomin learning aims to learn a representation with high utility while being uninformative about a specified target, with the latter achieved by minimising the mutual information between the representation and the target. It has broad applications, ranging from training fair prediction models against protected attributes, to unsupervised learning with disentangled representations. Recent works on infomin learning mainly use adversarial training, which involves training a neural network to estimate mutual information or its proxy and thus is slow and difficult to optimise. Drawing on recent advances in slicing techniques, we propose a new infomin learning approach, which uses a novel proxy metric to mutual information. We further derive an accurate and analytically computable approximation to this proxy metric, thereby removing the need of constructing neural network-based mutual information estimators. Experiments on algorithmic fairness, disentangled representation learning and domain adaptation verify that our method can effectively remove unwanted information with limited time budget.

## 1 Introduction

Learning representations that are uninformative about some target but still useful for downstream applications is an important task in machine learning with many applications in areas including algorithmic fairness [1, 2, 3, 4], disentangled representation learning [5, 6, 7, 8], information bottleneck [9, 10], and invariant representation learning [11, 12, 13, 14].

A popular method for the above task is adversarial training [1, 3, 4, 7, 15, 2, 11], where two neural networks, namely the encoder and the adversary, are trained jointly to compete with each other. The encoder's goal is to learn a representation with high utility but contains no information about the target. The adversary, on the contrary, tries to recover the information about the target from the learned representation as much as possible. This leads to a minmax game similar to that in generative adversarial networks [16]. Adversarial training is effective with a strong adversary, however, it is often challenging to train the adversary thoroughly in practice, due to time constraints and/or optimisation difficulties [17, 18]. In fact, recent studies have revealed that adversarial approaches may not faithfully produce an infomin representation in some cases [19, 20, 18, 21, 22]. This motivates us to seek a good, adversarial training-free alternative for scalable infomin learning.

In this work, we propose a new method for infomin learning which is almost as powerful as using a strong adversary but is highly scalable. Our method is inspired by recent advances in information theory which proposes to estimate mutual information in the sliced space [23]. We highlight the following contributions:

- We show that for infomin learning, an accurate estimate of mutual information (or its bound) is unnecessary: testing and optimising statistical independence in some sliced spaces is sufficient;

- We develop an analytical approximation to such sliced independence test, along with a scalable algorithm for infomin learning based on this approximation. No adversarial training is required.

36th Conference on Neural Information Processing Systems (NeurIPS 2022).

Importantly, the proposed method can be applied to a wide range of infomin learning tasks without any constraint on the form of variables or any assumption about the distributions. This contrasts our method to other adversarial training-free methods which are either tailored for discrete or univariate variables [24, 25, 25, 26, 22] or rely on variational approximation to distributions [10, 27, 19, 12].

## 2   Background

**Infomin representation learning**. Let $X \in \mathbb{R}^D$ be the data, $Y \in \mathbb{R}^{D'}$ be the target we want to predict from $X$. The task we consider here is to learn some representation $Z = f(X)$ that is useful for predicting $Y$ but is uninformative about some target $T \in \mathbb{R}^d$. Formally, this can be written as

$$\min_f \mathcal{L}(f(X); Y) + \beta \cdot I(f(X); T) \tag{1}$$

where $f$ is an encoder, $\mathcal{L}$ is some loss function quantifying the utility of $Z$ for predicting $Y$ and $I(f(X); T)$ quantifies the amount of information left in $Z$ about $T$. $\beta$ controls the trade-off between utility and uninformativeness. Many tasks in machine learning can be seen as special cases of this objective. For example, by setting $T$ to be (a set of) sensitive attributes e.g. race, gender or age, we arrive at fair representation learning [28, 1, 2, 21]. When using a stochastic encoder, by setting $Y$ to be $X$ and $T$ to be some generative factors e.g., a class label, we arrive at disentangled representation learning [5, 6, 7, 8]. Similarly, the information bottleneck method [9, 10] corresponds to setting $T = X$, which learns representations expressive for predicting $Y$ while being compressive about $X$.

**Adversarial training for infomin learning**. A key ingredient in objective (1) is to quantify $I(f(X); T)$ as the informativeness between $f(X)$ and $T$. One solution is to train a predictor for $T$ from $f(X)$ and use the prediction error as a measure of $I(f(X); T)$ [1, 3, 4, 29]. Another approach is to first train a classifier to distinguish between samples from $p(Z, T)$ vs. $p(Z)p(T)$ [7, 15] or to distinguish samples $Z \sim p(Z|T)$ with different $T$ [2, 11], then use the classification error to quantify $I(f(X); T)$. All these methods involve the training of a neural network $t$ to provide a lower-bound estimate to $I(f(X); T)$, yielding a minmax optimisation problem

$$\min_f \max_t \mathcal{L}(f(X); Y) + \beta \cdot \hat{I}_t(f(X); T) \tag{2}$$

where $\hat{I}_t(f(X); T)$ is an estimator constructed using $t$ that lower-bounds $I(f(X); T)$. The time complexity of optimising (2) is $O(L_1 L_2)$ where $L_1$ and $L_2$ are the number of gradient steps for the min and the max step respectively. The strength of $t$ is crucial for the quality of the learned representation [19, 20, 18, 21, 22]. For a strong adversary, a large $L_2$ is possibly needed, but this means a long training time. Conversely, a weak adversary may not produce a truly infomin representation.

## 3   Methodology

We propose an alternative to adversarial training for optimising (1). Our idea is to learn representation by the following objective, which replaces $I(f(X); T)$ in objective (1) with its 'sliced' version:

$$\min_f \mathcal{L}(f(X); Y) + \beta \cdot SI(f(X); T), \tag{3}$$

where $SI$ denotes the sliced mutual information, which was also considered in [23]. Informally, $SI$ is a 'facet' of mutual information that is much easier to estimate (ideally has closed form) but can still to some extent reflect the dependence between $Z$ and $T$. Optimising (3) is then equivalent to testing and minimising the dependence between $Z$ and $T$ from one facet. Importantly, while testing dependence through only a single facet may be insufficient, by testing and minimising dependence through various facets across a large number of mini-batches we eventually see $I(Z; T) \to 0$.

We show one instance for realising $SI$ whose empirical approximation $\hat{SI}$ has an analytic expression. The core of our method is Theorem 1, which is inspired by [4, 23].

**Theorem 1.** Let $Z \in \mathbb{R}^D$ and $T \in \mathbb{R}^d$ be two random variables that have moments. $Z$ and $T$ are statistically independent if and only if $SI(Z, T) = 0$ where $SI(Z, T)$ is defined as follows

$$SI(Z; T) = \sup_{h, g, \theta, \phi} \rho(h(\theta^\top Z), g(\phi^\top T)), \tag{4}$$

where $\rho$ is the Pearson correlation, $h, g : \mathbb{R} \to \mathbb{R}$ are Borel-measurable non-constant functions, and $\theta \in \mathbb{S}^{D-1}, \phi \in \mathbb{S}^{d-1}$ are vectors on the surfaces on $D$-dimensional and $d$-dimensional hyperspheres.

*Proof.* See the Appendix. □

We sketch here how this result relates to [4, 23]. [23] considers $\overline{SI}(Z;T)$, defined as the expected mutual information $\mathbb{E}[I(\theta^\top Z, \phi^\top T)]$ of slices of $Z$ and $T$, where the expectation is taken over respective Haar measures $\theta \in \mathbb{S}^{D-1}$, $\phi \in \mathbb{S}^{d-1}$. Instead of considering the mutual information $I(\theta^\top Z, \phi^\top T)$ in the average case, we handle Pearson correlation over the supreme functions $h, g$ as defined above, which links to Rényi's maximal correlation [4, 30, 31, 32] and has some interesting properties suitable for infomin representation learning (explained later).

We call $\theta$ and $\phi$ the slices for $Z$ and $T$ respectively, and $\theta^\top Z$, $\phi^\top T$ the sliced $Z$ and $T$ respectively.

Intuitively, Theorem 1 says that in order to achieve $I(Z;T) \to 0$, we need not estimate $I(Z;T)$ in the original space; rather we can test (and optimise) independence in the sliced space as realised by (4). Note other realisations of the sliced mutual information $SI$ can also be used. The major merit of the realisation (4) is that (a) it is bounded in $[0, 1]$, which eases hyperparameter tuning and makes learning stable; (b) it allows an analytic expression for its empirical approximation, as shown below.

**Analytic approximation to *SI*.** An empirical approximation to (4) is

$$SI(Z;T) \approx \sup_{i,j} \sup_{h_i, g_j} \rho(h_i(\theta_i^\top Z), g_j(\phi_j^\top T)),$$

$$\text{where} \quad \theta_i \sim \mathcal{U}(\mathbb{S}^{D-1}), \ i = 1, ..., S, \qquad \phi_j \sim \mathcal{U}(\mathbb{S}^{d-1}), \ j = 1, ..., S. \quad (5)$$

i.e., we approximate (4) by randomly sampling a number of slices $\theta, \phi$ uniformly from the surface of two hyperspheres $\mathbb{S}^{D-1}$ and $\mathbb{S}^{d-1}$ and pick those slices where the sliced $Z$ and the sliced $T$ are maximally associated. With a large number of slices, it is expected that (5) will approximate (4) well. We refer to [23] for a theoretical analysis on the number of required slices in estimator-agnostic settings. In Appendix B we also investigate empirically how this number will affect performance.

For each slicing direction, we further assume that the supreme functions $h_i, g_j : \mathbb{R} \to \mathbb{R}$ for that direction can be well approximated by $K$-order polynomials given sufficiently large $K$, i.e.

$$h_i(a) \approx \hat{h}_i(a) = \sum_{k=0}^{K} w_{ik}\sigma(a)^k, \qquad g_j(a) \approx \hat{g}_j(a) = \sum_{k=0}^{K} v_{jk}\sigma(a)^k,$$

where $\sigma(\cdot)$ is a monotonic function which maps the input to the range of $[-1, 1]$. Its role is to ensure that $\sigma(a)$ always has finite moments, so that the polynomial approximation is well-behaved. Note no information is lost by applying $\sigma(\cdot)$. Here we take $\sigma(\cdot)$ as the tanh function. Other approximations can also be used (e.g. those based on random features [33]), which can be explored in the future.

One may wonder if ignoring the higher-order terms $\sigma(a)^{k'}, k' > K$ will cause inaccuracy in dependence modelling. In Appendix A we show that if $|\hat{h}_i(a) - h_i(a)| \le \epsilon$ and $|\hat{g}_j(a) - g_j(a)| \le \epsilon$ for $\forall i, j, a$, then as $\epsilon \to 0$ we have $|\rho_{ij}^* - \hat{\rho}_{ij}^*| = o(\epsilon)$ where $\rho_{ij}^* = \rho(h_i(\theta_i^\top Z), g_j(\phi_j^\top T))$ is the true correlation and $\hat{\rho}_{ij}^*$ is its $K$-order approximation. In such case we have $\hat{\rho}_{ij}^* \approx 0$ whenever $\rho_{ij}^* \approx 0$.

With this polynomial approximation, the solving of $h_i, g_j$ reduces to finding their weights $w_i, v_j$:

$$w_i, v_j = \arg\max_{w,v} \rho(w^\top Z_i', v^\top T_j'),$$

$$Z_i' = [\sigma(\theta_i^\top Z), ..., \sigma(\theta_i^\top Z)^K], \qquad T_j' = [\sigma(\phi_j^\top T), ..., \sigma(\phi_j^\top T)^K]$$

This is known as canonical correlation analysis [34] and can be solved analytically by eigendecomposition. Hence we can find the weights for all pairs of $h_i, g_j$ by doing $S^2$ eigendecompositions.

In fact, the functions $h_i, g_j$ for all $i, j$ can be solved simultaneously by performing a larger eigendecomposition only once. We do this by finding $w, v$ that maximise the following quantity:

$$\hat{SI}_{\Theta,\Phi}(Z;T) = \sup_{w,v} \rho(w^\top Z', v^\top T'), \quad (6)$$

$$\text{where} \quad Z_i' = [\sigma(\theta_i^\top Z), ..., \sigma(\theta_i^\top Z)^K], \qquad T_j' = [\sigma(\phi_j^\top T), ..., \sigma(\phi_j^\top T)^K]$$

$$\theta_i \sim \mathcal{U}(\mathbb{S}^{D-1}), \ i = 1, ..., S, \qquad \phi_j \sim \mathcal{U}(\mathbb{S}^{d-1}), \ j = 1, ..., S.$$

The benefits for solving $f_i, g_j$ for all slices jointly are two-fold. The first is better computational efficiency in practice, as it avoids invoking a for loop and has better affinity to modern deep learning

infrastructure and libraries (e.g. Tensorflow [35] and PyTorch [36]) which are optimised for matrix-based operations. We highlight that this joint-solving strategy will not violate our original objective (i.e. minimising (5)), as backed up by Theorem 2 below. The intuition behind Theorem 2 is that if all $Z'_i$ and $T'_j$ together cannot achieve a high correlation, they alone can not either:

**Theorem 2.** Provided that each $h_i, g_j$ in (5) are $K$-order polynomials , given the sampled $\Theta = \{\theta_i\}_{i=1}^S, \Phi = \{\phi_j\}_{j=1}^S$, we have $\hat{SI}_{\Theta,\Phi}(Z;T) \leq \epsilon \Rightarrow \sup_{i,j} \sup_{h_i,g_j} \rho(h_i(\theta_i^\top Z), g_j(\phi_j^\top T)) \leq \epsilon$.

*Proof.* See Appendix A. □

Theorem 2 says that the solution of (6) yields an upper bound of (5). This essentially means that we are safe to replace (5) with (6). In addition to computational efficiency, another benefit for solving $g_j, h_j$ jointly is stronger power in independence testing. More specifically, it allows us to unify the power of different slices: while some sliced directions may be weak for detecting dependence, they together as a whole (by treating them as a 'big slice' [37]) can compensate for each other, yielding a more powerful test. For these reasons, we use $\hat{SI}_{\Theta,\Phi}(Z;T)$ as the approximation to $SI(Z;T)$.

---

| **Algorithm 1** Adversarial Infomin Learning | **Algorithm 2** Slice Infomin Learning |
|---|---|
| **Input:** data $\mathcal{D} = \{X^{(n)}, Y^{(n)}, T^{(n)}\}_{n=1}^N$ | **Input:** data $\mathcal{D} = \{X^{(n)}, Y^{(n)}, T^{(n)}\}_{n=1}^N$ |
| **Output:** $Z = f(X)$ that optimises (1) | **Output:** $Z = f(X)$ that optimises (1) |
| **Hyperparams:** $\beta, N', L_1, L_2$ | **Hyperparams:** $\beta, N', L, S$ |
| **Parameters:** encoder $f$, MI estimator $t$ | **Parameters:** encoder $f$, weights $w, v$ in $\hat{SI}$ |
| | |
| **for** $l_1$ in 1 to $L_1$ **do** | **for** $l$ in 1 to $L$ **do** |
|   sample mini-batch $\mathcal{B}$ from $\mathcal{D}$ |   sample mini-batch $\mathcal{B}$ from $\mathcal{D}$ |
|   sample $\mathcal{D}'$ from $\mathcal{D}$ whose size $N' < N$ |   sample $\mathcal{D}'$ from $\mathcal{D}$ whose size $N' < N$ |
|   ▷ *Max-step* |   ▷ *Max-step* |
|   **for** $l_2$ in 1 to $L_2$ **do** |   sample $S$ slices $\Theta = \{\theta_i\}_{i=1}^S, \Phi = \{\phi_j\}_{j=1}^S$ |
|     $t \leftarrow t + \eta\nabla_t \hat{I}_t(f(X); T)$ with data in $\mathcal{D}'$ |   solve the parameters $w, v$ in $\hat{SI}$ analytically |
|   **end for** |   with $\Theta, \Phi, \mathcal{D}'$ by eigendecomposition |
|   ▷ *Min-step* |   ▷ *Min-step* |
|   $f \leftarrow f - \eta\nabla_f[\mathcal{L}(f(X); Y) + \beta\hat{I}_t(f(X); T)]$ |   $f \leftarrow f - \eta\nabla_f[\mathcal{L}(f(X); Y) + \beta\hat{SI}(f(X); T)]$ |
|   with data in $\mathcal{B}$ |   with data in $\mathcal{B}$ |
| **end for** | **end for** |
| **return** $Z = f(X)$ | **return** $Z = f(X)$ |

---

**Mini-batch learning algorithm**. Given the above approximation (6) to $SI$, we can now elaborate the details of mini-batch learning. For each mini-batch $\mathcal{B}$, we execute the following steps:

- *Max-step.* Sample $S$ slices $\Theta = \{\theta_i\}_{i=1}^S, \Phi = \{\phi_j\}_{j=1}^S$ and a subset of the data $\mathcal{D}' \subset \mathcal{D}$ (here $|\mathcal{D}'|$ can be larger than $|\mathcal{B}|$). Learn the weights $w, v$ of (6) with $\Theta, \Phi, \mathcal{D}'$ by eigendecomposition;

- *Min-step.* Set $\hat{SI}(Z, T) = \rho(w^\top Z', v^\top T')$ with $w, v$ solved in the max-step and $Z', T'$ defined in (6). Update $f$ by SGD: $f \leftarrow f - \eta\nabla_f[\mathcal{L}(f(X), Y) + \beta\hat{SI}(f(X), T)]$ with the mini-batch data.

The whole learning procedure is shown in Algorithm 2. Compared to that of adversarial methods [2, 11, 1, 3, 21, 4] as shown in Algorithm 1, we replace the optimisation of neural net in the max step with an analytical eigendecomposition step. The time complexity of eigendecomposition is $O(S^3)$.

As an optional strategy, during mini-batch learning, one may actively seek more informative slices for independence testing by optimising the sampled slices with a few gradient steps (e.g. 2-5):

$$\Theta \leftarrow \Theta - \xi\nabla_\Theta \hat{SI}_{\Theta,\Phi}(Z, T), \qquad \Phi \leftarrow \Phi - \xi\nabla_\Phi \hat{SI}_{\Theta,\Phi}(Z, T) \qquad (7)$$

which is still very cheap to execute. Such a strategy is may be useful when most of the sampled slices are weak in detecting dependence, which typically happens in later iterations of learning where $I(Z; T) \approx 0$. We can activate such strategy whenever the estimated $SI$ is very low (e.g. $\hat{SI}(Z; T) < 0.05$). Since the optimisation of slices only happens in late learning iterations, and we only apply a small number of gradient steps, the overall execution time will only increase negligibly.

# 4 Related works

**Neural mutual information estimators**. A set of neural network-based methods [38, 39, 40, 41] have been proposed to estimate the mutual information (MI) between two random variables, most of which work by maximising a lower bound of MI [42]. These neural MI estimators are in general more powerful than non-parametric methods [43, 44, 45, 46] when trained thoroughly, yet the time spent on training may become the computational bottleneck when applied to infomin learning.

**Upper bound for mutual information**. Another line of method for realising the goal of infomin learning without adversarial training is to find an upper bound for mutual information [10, 19, 12, 47, 48]. However, unlike lower bound estimate, upper bound often requires knowledge of either the conditional densities or the marginal densities [42] which are generally not available in practice. As such, most of these methods introduce a variational approximation to these densities whose choice/estimate may be difficult. Our method on the contrary needs not to approximate any densities.

**Slicing techniques**. A series of successes have been witnessed for the use of slicing methods in machine learning and statistics [49], with applications in generative modelling [50, 51, 52, 53], statistical test [54] and mutual information estimate [23]. Among them, the work [23] who proposes the concept of sliced mutual information is very related to this work and directly inspires our method. Our contribution is a novel realisation of sliced mutual information suitable for infomin learning.

**Fair machine learning**. One application of our method is to encourage the fairness of a predictor. Much efforts have been devoted for the same purpose, however most of the existing methods can either only work at the classifier level [25, 24, 4, 55], or only focus on the case where the sensitive attribute is discrete or univariate [22, 56, 26, 55, 28], or require adversarial training [2, 11, 1, 3, 21, 4]. Our method on the contrary has no restriction on the form of the sensitive attribute, can be used in both representation level and classifier level, and require no adversarial training of neural networks.

**Disentangled representation learning**. Most of the methods in this field work by penalising the discrepancy between the joint distribution $P = q(Z)$ and the product of marginals $Q = \prod_d^D q(Z_d)$ [6, 57, 27, 58, 7].[1] However, such discrepancy is often non-trivial to estimate, so one has to resort to Monte Carlo estimate ($\beta$-TCVAE [27]), to train a neural network estimator (FactorVAE [7]) or to assess the discrepancy between $P$ and $Q$ by only their moments (DIP-VAE [57]). Our method avoids assessing distribution discrepancy directly and instead perform independence test in the sliced space.

# 5 Experiments

We evaluate our approach on four tasks: independence testing, algorithmic fairness, disentangled representation learning, domain adaptation. Code is available at `github.com/cyz-ai/infomin`.

**Evaluation metric**. To assess how much information is left in the learned representation $Z \in \mathbb{R}^D$ about the target $T \in \mathbb{R}^K$, we calculate the Rényi's maximal correlation $\rho^*(Z, T)$ between $Z$ and $T$:

$$\rho^*(Z, T) = \sup_{h,g} \rho(h(Z), g(T)) \tag{8}$$

which has the properties $\rho^*(Z, T) = 0$ if and only if $Z \perp T$ and $\rho^*(Z, T) = 1$ if $h(Z) = g(T)$ for some deterministic functions $h, g$ [30]. One can also understand this metric as the easiness of predicting (the transformed) $T$ from $Z$, or vice versa.[2] As there is no analytic solution for the supremum in (8), we approximate them by two neural networks $h, g$ trained with SGD. Early stopping and dropout are applied to avoid overfitting. The reliability of this neural approximation has been verified by the literature [4] and is also confirmed by our experiments; see Appendix B.

This metric is closely related to existing metrics/losses used in fairness and disentangled representation learning such as demographic parity (DP) [24] and total correlation (TC) [7]. For example, if $\rho^*(Z, T) \to 0$ then it is guaranteed that $\hat{Y} \perp T$ for any predictor $\hat{Y} = F(Z)$, so $\rho^*(Z, T)$ is an upper bound for DP. Similarly, $\rho^*(Z, T)$ coincides with TC which also assesses whether $Z \perp T$. In additional to this metric, we will also use some task-specific metric; see each experiment below.

**Baselines**. We compare the proposed method (denoted as "Slice") with the following approaches:

---

[1]Note there exist methods based on group theory [59, 60, 61] which do not assess distribution discrepancy.

[2]It can be shown $\rho^*(Z, T)$ is equivalent to the normalised mean square error between $h(Z)$ and $g(T)$.

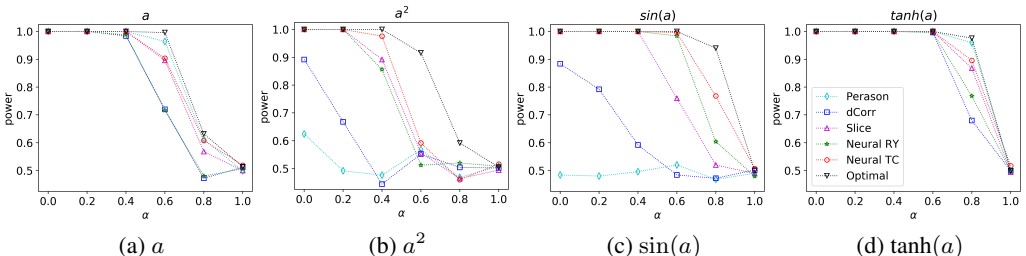

Figure 1: Comparison of the test power of different independence test methods. The x-axis corresponds to different values for the dependence level $\alpha$ and the y-axis corresponds to the test power.

- *Pearson*, which quantifies $I(Z;T)$ by the Pearson correlation coefficient $\frac{1}{DK}\sum_d^D\sum_k^K\rho(Z_d;T_k)$. It was used in [5, 57] as an easy-to-compute proxy to MI to learn disentangled representations;

- *dCorr*, i.e. distance correlation, a non-parametric method for the quantifying the independence between two vectors [46]. It was applied in [41] as a surrogate to MI for representation learning;

- *Neural Rényi*, an adversarial method for fair machine learning [25] which quantifies $I(Z;T)$ by the Rényi correlation $\rho^*(Z,T) = \sup_{h,g}\rho(h(Z),g(T))$ with $h,g$ approximated by neural networks. It can be seen as training a predictor to predict (the transformed) $T$ from $Z$ and is closely related to many existing methods in algorithmic fairness and domain adaptation [1, 2, 3, 4, 29];

- *Neural TC*, an adversarial method for learning disentangled representation [7, 62] which quantifies $I(Z;T)$ by the total correlation $TC(Z,T) = KL[p(Z,T)\|p(Z)p(T)]$. To computes TC, a classifier is trained to classify samples from $p(Z,T)$ and samples from $p(Z)p(T)$. This method can also be seen as a variant of the popular MINE method [38] for mutual information estimate.

- *v-CLUB*, i.e. variational contrastive log upper bound, which introduces a (learnable) variational distribution $q(T|Z)$ to form an upper bound of MI [47]: $I(Z;T) \leq \mathbb{E}_{p(Z,T)}[\log q(T|Z)] - \mathbb{E}_{p(Z)p(T)}[\log q(T|Z)]$. Like adversarial method, $q(T|Z)$ can be learned by a few gradient steps.

For a fair comparison, for adversarial training-based approaches (i.e. Neural Rényi, Neural TC), we ensure that the training time of the neural networks in these methods is at least the same as the execution time of our method or longer. We do this by controlling the number of adversarial steps $L_2$ in Algorithm 1. The same setup is used for v-CLUB. See each experiment for the detailed time.

**Hyperparameter settings**. Throughout our experiments, we use 200 slices. We find that this setting is robust across different tasks. An ablation study on the number of slices is given in Appendix B. The order of the polynomial used in (6) namely $K$ is set as $K = 3$ and is fixed across different tasks.

**Computational resource**. All experiments are done with a single NVIDIA GeForce Tesla T4 GPU.

## 5.1 Independence testing

We first verify the efficacy of our method as a light-weight but powerful independence test. For this purpose, we investigate the test power of the proposed method on various synthetic dataset with different association patterns between two random variables $X \in \mathbb{R}^{10}, Y \in \mathbb{R}^{10}$ and compared to that of the baselines. The test power is defined as the ability to discern samples from the joint distribution $p(X,Y)$ and samples from the product of marginal $p(X)p(Y)$ and is expressed as a probability $p \in [0,1]$. The data is generated as $Y = (1-\alpha)\langle t(\mathbf{A}X)\rangle + \alpha\epsilon, X_d \sim \mathcal{U}[-3,3], \mathbf{A}_{dd} = 1, \mathbf{A}_{dk} = 0.2, \epsilon \sim \mathcal{N}(\epsilon;\mathbf{0},\mathbf{I}), \alpha \in (0,1)$ and $\langle\cdot\rangle$ is a scaling operation that scales the operand to the range of $[0,1]$ according to the minimum and maximum values in the population. The function $t(\cdot)$ determines the association pattern between $X$ and $Y$ and is chosen from one of the following: $t(a) = a, a^2, \sin(a), \tanh(a)$. The factor $\alpha$ controls the strength of dependence between $X$ and $Y$.

All tests are done on 100 samples and are repeated 1,000 times. We choose this sample number as it is a typical batch size in mini-batch learning. For methods involving the learning of parameters (i.e. Slice, Neural Rényi, Neural TC), we learn their parameters from 10,000 samples. The time for learning the parameters of Slice, Neural Rényi and Neural TC are 0.14s, 14.37s, 30.18s respectively. For completeness, we also compare with the 'optimal test' which calculates the Rényi correlation $\rho^*(X,Y) = \rho(h(X),g(Y))$ with the functions $h,g$ exactly the same as the data generating process.

Table 1: Learning fair representations on the US Census Demographic dataset. Here the utility of the representation is measured by $\rho^*(Z,Y)$, while $\rho^*(Z,T)$ is used to quantify the fairness of the representation. Training time is also provided as the seconds required per max step.

|  | N/A | Pearson | dCorr | Slice | Neural Rényi | Neural TC | vCLUB |
|---|---|---|---|---|---|---|---|
| $\rho^*(Z,Y)\uparrow$ | $0.95 \pm 0.00$ | $0.95 \pm 0.00$ | $0.95 \pm 0.00$ | $0.95 \pm 0.01$ | $0.95 \pm 0.01$ | $0.95 \pm 0.02$ | $0.94 \pm 0.02$ |
| $\rho^*(Z,T)\downarrow$ | $0.92 \pm 0.02$ | $0.84 \pm 0.08$ | $0.47 \pm 0.08$ | $0.07 \pm 0.02$ | $0.23 \pm 0.10$ | $0.27 \pm 0.03$ | $0.16 \pm 0.10$ |
| time (sec./max step) | 0.000 | 0.012 | 0.087 | 0.102 | 0.092 | 0.097 | 0.134 |

Table 2: Learning fair representations on the UCI Adult dataset. Here the utility of the representation is measured by $\rho^*(Z,Y)$, while $\rho^*(Z,T)$ is used to quantify the fairness of the representation.

|  | N/A | Pearson | dCorr | Slice | Neural Rényi | Neural TC | vCLUB |
|---|---|---|---|---|---|---|---|
| $\rho^*(Z,Y)\uparrow$ | $0.99 \pm 0.00$ | $0.99 \pm 0.00$ | $0.97 \pm 0.01$ | $0.98 \pm 0.01$ | $0.97 \pm 0.01$ | $0.98 \pm 0.02$ | $0.97 \pm 0.02$ |
| $\rho^*(Z,T)\downarrow$ | $0.94 \pm 0.02$ | $0.91 \pm 0.06$ | $0.71 \pm 0.06$ | $0.08 \pm 0.02$ | $0.17 \pm 0.08$ | $0.36 \pm 0.13$ | $0.26 \pm 0.12$ |
| time (sec./max step) | 0.000 | 0.015 | 0.071 | 0.112 | 0.107 | 0.131 | 0.132 |

Figure 1 shows the power of different methods under various association patterns $t$ and dependence levels $\alpha$. Overall, we see that the proposed method can effectively detect dependence in all cases, and has a test power comparable to neural network-based methods. Non-parametric tests, by contrast, fail to detect dependence in quadratic and periodic cases, possibly due to insufficient power. Neural TC is the most powerful test among all the methods considered, yet it requires the longest time to train. We also see that the proposed method is relatively less powerful when $\alpha \geq 0.8$, but in such cases the statistical dependence between $X$ and $Y$ is indeed very weak (also see Appendix B). The results suggest that our slice method can provide effective training signals for infomin learning tasks.

## 5.2 Algorithmic fairness

For this task, we aim to learn fair representations $Z \in \mathbb{R}^{80}$ that are minimally informative about some sensitive attribute $T$. We quantify how sensitive $Z$ is w.r.t $T$ by Rényi correlation $\rho^*(Z,T)$ calculated using two neural nets. Smaller $\rho^*(Z,T)$ is better. The utility of the learned representation i.e., $\mathcal{L}(Z;Y)$ is quantified by $\rho^*(Z,Y)$. This formulation for utility, as aforementioned, is equivalent to measuring how well we can predict $Y$ from $Z$. In summary, the learning objective is:

$$\max \rho^*(Z;Y) - \beta \hat{I}(Z;T),$$

where $\hat{I}(Z;T)$ is estimated by the methods mentioned above. For each dataset considered, we use 20,000 data for training and 5,000 data for testing respectively. We carefully tune the hyperparameter $\beta$ for each method so that the utility $\rho^*(Z;Y)$ of that method is close to that of the plain model (i.e. the model trained with $\beta = 0$, denoted as N/A below; other experiments below have the same setup). For all methods, we use $5,000$ samples in the max step (so $N' = 5,000$ in Algorithm 1, 2).

**US Census Demographic data**. This dataset is an extraction of the 2015 American Community Survey, with 37 features about 74,000 census tracts. The target $Y$ to predict is the percentage of children below poverty line in a tract, and the sensitive attribute $T$ is the ratio of women in that tract. The result is shown in Table 1. From the table we see that the proposed slice method produces highly fair representation with good utility. The low $\rho^*(Z,T)$ value indicates that it is unlikely to predict $T$ from $Z$ in our method. While adversarial methods can also to some extent achieve fairness, it is still not comparable to our method, possibly because the allocated training time is insufficient (in Appendix B we study how the effect of the training time). Non-parameteric methods can not produce truly fair representation, despite they are fast to execute. v-CLUB, which estimates an upper bound of MI, achieves better fairness than adversarial methods on average, but has a higher variance [63].

**UCI Adult data**. This dataset contains census data for 48,842 instances, with 14 attributes describing their education background, age, race, marital status, etc. Here, the target $Y$ to predict is whether the income of an instance is higher than 50,000 USD, and the sensitive attribute $T$ is the race group. The result is summarised in Table 2. Again, we see that the proposed slice method outperforms other methods in terms of both fairness and utility. For this dataset, Neural Rényi also achieves good fairness, although the gap to our method is still large. Neural TC, by contrast, can not achieve a

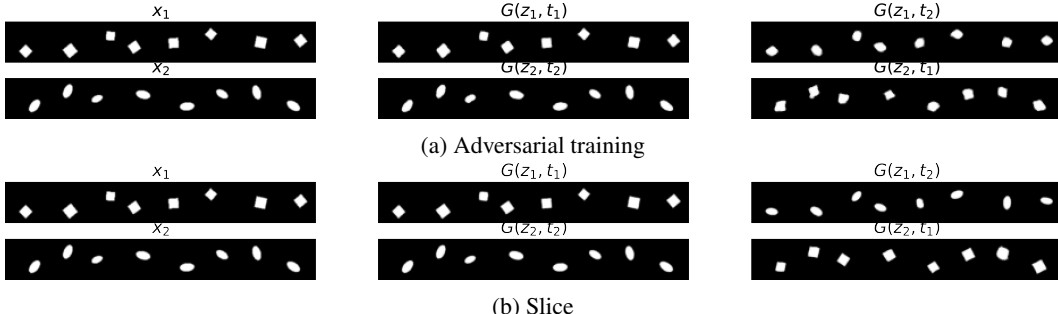

(a) Adversarial training

(b) Slice

Figure 2: Label swapping experiments on Dsprite dataset. Left: the original image $X$. Middle: reconstructing $X \approx G(Z, T)$ using $Z = E(X)$ and the true label $T$. Right: reconstructing $X' = G(Z, T')$ using $Z = E(X)$ and a swapped label $T' \neq T$. Changing $T$ should only affects the style.

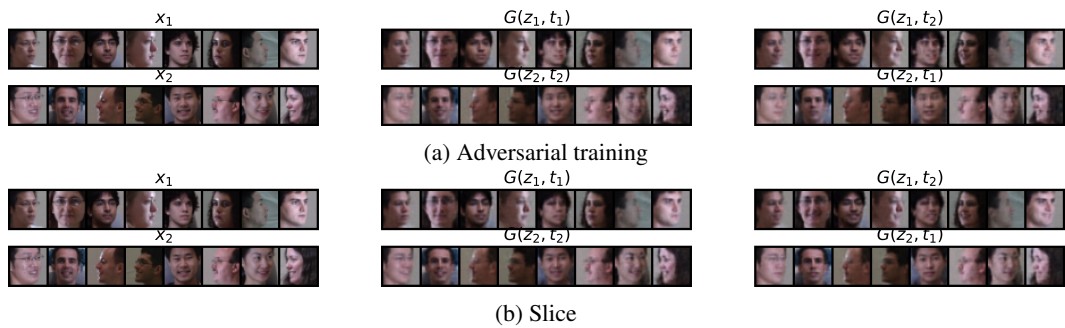

(a) Adversarial training

(b) Slice

Figure 3: Label swapping experiments on CMU-PIE dataset. Left: the original image $X$. Middle: reconstructing $X \approx G(Z, T)$ using $Z = E(X)$ and the true label $T$. Right: reconstructing $X' = G(Z, T')$ using $Z = E(X)$ and a swapped label $T' \neq T$. Changing $T$ only affect the expression.

comparable level of fairness under the time budget given — a phenomenon also observed in the US Census dataset. This is possibly because the networks in Neural TC require longer time to train. The v-CLUB method does not work very satisfactorily on this task, possibly because the time allocated to learn the variational distribution $q(Z|T)$ is not enough, leading to a loose upper bound of $I(Z; T)$.

## 5.3 Disentangled representation learning

We next apply our method to the task of disentangled representation learning, where we wish to discover some latent generative factors irrelevant to the class label $T$. Here, we train a conditional autoencoder $X \approx G(Z, T)$ to learn representation $Z = E(X)$ which encodes label-irrelevant information of $X$. The target to recover is $Y = X$. The utility of $Z$ is therefore quantified as the reconstruction error: $\mathcal{L}(Z; Y) = \mathbb{E}[\|G(Z, T) - X\|_2^2]$, resulting in the following learning objective:

$$\max \mathbb{E}[\|G(Z, T) - X\|_2^2] - \beta \hat{I}(Z; T).$$

The conditional autoencoder uses a architecture similar to that of a convolutional GAN [64], with the difference being that we insert an adaption layer $Z' = \text{MLP}(Z, T)$ before feeding the features to the decoder. See Appendix B for the details of its architecture. All images are resized to $32 \times 32$. For all methods, we use $10,000$ samples in the max step (so $N' = 10,000$ in Algorithms 1 and 2).

**Dsprite**. A 2D shape dataset [65] where each image is generated by four latent factors: shape, rotation, scale, locations. Here the class label $T$ is the shape, which ranges from (square, ellipse, heart). For this dataset, we train the autoencoder 100 iterations with a batch size of 512. The dimensionality of the representation for this task is 20 i.e. $Z \in \mathbb{R}^{20}$. As in the previous experiments, we provide quantitative comparisons of the utility and disentanglement of different methods in Table 3. In addition, we provide a qualitative comparison in Figure 2 which visualises the original image, the reconstructed image and the reconstructed image with a swapped label. From Table 3, we see that the proposed method achieve very low $\rho^*(Z, T)$ while maintaining good MSE, suggesting that we

Table 3: Learning label-irrelevant representations on the Dsprite dataset. Here the utility of the representation is measured by MSE, while $\rho^*(Z, T)$ is used to quantify the level of disentanglement of the representation. Training time is also provided as the seconds needed per max step. $\text{Acc}(\hat{T})$ is the accuracy trying to predict $T$ from $Z$. As there are 3 classes, the ideal value for $\text{Acc}(\hat{T})$ is $0.33^*$.

| | N/A | Pearson | dCorr | Slice | Neural Rényi | Neural TC | vCLUB |
|---|---|---|---|---|---|---|---|
| MSE ↓ | $0.37 \pm 0.01$ | $0.44 \pm 0.02$ | $0.55 \pm 0.03$ | $0.50 \pm 0.01$ | $0.61 \pm 0.04$ | $0.49 \pm 0.03$ | $0.65 \pm 0.04$ |
| $\rho^*(Z, T) \downarrow$ | $0.91 \pm 0.03$ | $0.81 \pm 0.07$ | $0.62 \pm 0.07$ | $0.08 \pm 0.02$ | $0.48 \pm 0.05$ | $0.34 \pm 0.06$ | $0.22 \pm 0.08$ |
| $\text{Acc}(\hat{T})$ | $0.98 \pm 0.01$ | $0.89 \pm 0.03$ | $0.76 \pm 0.05$ | $0.32 \pm 0.02$ | $0.55 \pm 0.04$ | $0.54 \pm 0.04$ | $0.48 \pm 0.03$ |
| time (sec./max step) | 0.000 | 0.201 | 0.412 | 0.602 | 0.791 | 0.812 | 0.689 |

Table 4: Learning label-irrelevant representations on the CMU-PIE dataset. Here the utility of the representation is measured by MSE, while $\rho^*(Z, T)$ is used to quantify the level of disentanglement of the representation. Training time is also provided as the seconds needed per max step. $\text{Acc}(\hat{T})$ is the accuracy trying to predict $T$ from $Z$. As there are 2 classes, the ideal value for $\text{Acc}(\hat{T})$ is $0.50^*$.

| | N/A | Pearson | dCorr | Slice | Neural Rényi | Neural TC | vCLUB |
|---|---|---|---|---|---|---|---|
| MSE ↓ | $1.81 \pm 0.04$ | $1.85 \pm 0.05$ | $2.08 \pm 0.08$ | $2.15 \pm 0.07$ | $2.46 \pm 0.06$ | $1.99 \pm 0.12$ | $2.02 \pm 0.10$ |
| $\rho^*(Z, T) \downarrow$ | $0.76 \pm 0.04$ | $0.55 \pm 0.03$ | $0.27 \pm 0.07$ | $0.07 \pm 0.01$ | $0.36 \pm 0.04$ | $0.39 \pm 0.06$ | $0.16 \pm 0.06$ |
| $\text{Acc}(\hat{T})$ | $0.91 \pm 0.00$ | $0.76 \pm 0.03$ | $0.71 \pm 0.06$ | $0.51 \pm 0.03$ | $0.73 \pm 0.03$ | $0.76 \pm 0.04$ | $0.68 \pm 0.05$ |
| time (sec./max step) | 0.000 | 0.184 | 0.332 | 0.581 | 0.750 | 0.841 | 0.621 |

*For the plain model, $\text{Acc}(\hat{T})$ is not necessarily around 1.0, as $Z$ needs not encode all content of the image.

may have discovered the true label-irrelevant generative factor for this dataset. This is confirmed visually by Figure 2(b), where by changing $T$ in reconstruction we only change the style. By contrast, the separation between $T$ and $Z$ is less evident in adversarial approach, as can be seen from Table 3 as well as from Figure 2(a) (see e.g. the reconstructed ellipses in the third column of the figure. They are more like a interpolation between ellipses and squares).

**CMU-PIE**. A colored face image dataset [66] where each face image has different pose, illumination and expression. We use its cropped version [67]. Here the class label $T$ is the expression, which ranges from (neutral, smile). We train an autoencoder with 200 iteration and a batch size of 128. The dimensionality of the representation for this task is 128 i.e. $Z \in \mathbb{R}^{128}$. Figure 3 and Table 4 shows the qualitative and quantitative results respectively. From Figure 3, we see that our method can well disentangle expression and non-expression representations: one can easily modify the expression of a reconstructed image by only changing $T$. Other visual factors of the image including pose, illumination, and identity remain the same after changing $T$. Adversarial approach can to some extent achieve disentanglement between $Z$ and $T$, however such disentanglement is imperfect: not all of the instances can change the expression by only modifying $T$. This is also confirmed quantitatively by Table 4, where one can see the relatively high $\rho^*(Z, T)$ values in adversarial methods. For this task, v-CLUB also achieves a low $\rho^*(Z, T)$ value, though it is still outperformed by our method.

## 5.4 Domain adaptation

We finally consider the task of domain adaptation, where we want to learn some representation $Z$ that can generalise across different datasets. For this task, a common assumption is that we have assess to two dataset $\mathcal{D}_s = \{X^{(i)}, Y^{(i)}\}_{i=1}^n$ and $\mathcal{D}_t = \{X^{(j)}\}_{j=1}^m$ whose classes are the same but are collected differently. Only the data in $\mathcal{D}_s$ has known labels. Following [47], we learn $Z$ as follows:

$$Z_c = f_c(X), \qquad Z_d = f_d(X)$$

$$\mathcal{L}_c = \mathbb{E}_{X,Y \in \mathcal{D}_s}[Y^\top \log C(Z_c)], \qquad \mathcal{L}_d = \mathbb{E}_{X \in \mathcal{D}_s}[\log D(Z_d)] + \mathbb{E}_{X \in \mathcal{D}_t}[\log(1 - D(Z_d))],$$

$$\min \mathcal{L}_c + \mathcal{L}_d + \beta \hat{I}(Z_c, Z_d),$$

where $Z_c, Z_d$ are disjoint parts of $Z$ that encode the content information and the domain information of $X$ separately. $C$ is the content classifier that maps $X$ to a $(K-1)$-simplex ($K$ is the number of classes) and $D$ is the domain classifier that distinguishes the domain from which $X$ comes. Since the

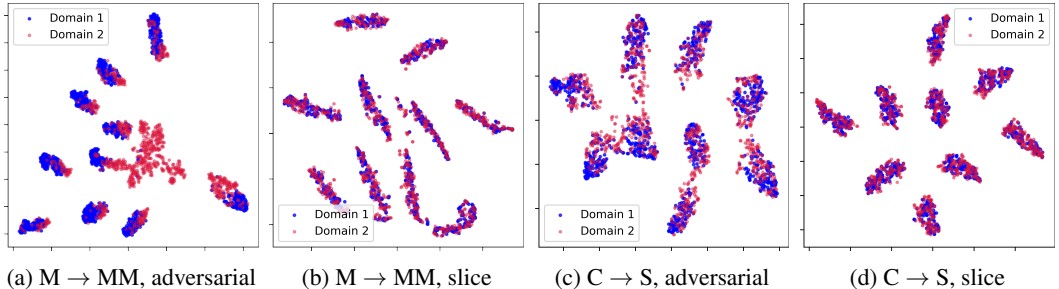

| (a) M → MM, adversarial | (b) M → MM, slice | (c) C → S, adversarial | (d) C → S, slice |

Figure 4: T-SNE plots of the learned content representations $Z_c$ in domain adaptation tasks. (a)(c) show the cases when the adversary is not trained thoroughly (i.e. $L_2$ in Algorithm 1 is set too small).

Table 5: Learning domain-invariant representations. Here $\text{Acc}(\hat{Y}_s)$ and $\text{Acc}(\hat{Y}_t)$ are the classification accuracy in the source and the target domains respectively. Time used per max step is given.

| | MNIST → MNIST-M | | | | | CIFAR10 → STL10 | | | |
|---|---|---|---|---|---|---|---|---|---|
| | N/A | Slice | Neural TC | vCLUB | | N/A | Slice | Neural TC | vCLUB |
| $\text{Acc}(\hat{Y}_s)\uparrow$ | $99.3 \pm 0.04$ | $99.2 \pm 0.02$ | $99.2 \pm 0.04$ | $99.0 \pm 0.03$ | $\text{Acc}(\hat{Y}_s)\uparrow$ | $93.0 \pm 0.03$ | $92.5 \pm 0.03$ | $92.4 \pm 0.03$ | $92.1 \pm 0.04$ |
| $\text{Acc}(\hat{Y}_t)\uparrow$ | $46.3 \pm 0.03$ | $98.5 \pm 0.45$ | $80.1 \pm 0.17$ | $93.8 \pm 0.10$ | $\text{Acc}(\hat{Y}_t)\uparrow$ | $75.9 \pm 0.09$ | $82.3 \pm 0.03$ | $80.8 \pm 0.08$ | $78.5 \pm 0.11$ |
| $\rho^*(Z_c, Z_d)\downarrow$ | $0.86 \pm 0.05$ | $0.06 \pm 0.01$ | $0.64 \pm 0.04$ | $0.49 \pm 0.12$ | $\rho^*(Z_c, Z_d)\downarrow$ | $0.43 \pm 0.05$ | $0.08 \pm 0.01$ | $0.39 \pm 0.07$ | $0.42 \pm 0.09$ |
| time (sec./step) | $0.000$ | $2.578$ | $3.282$ | $3.123$ | time (sec./step) | $0.000$ | $3.146$ | $3.222$ | $3.080$ |

classifier $C$ only sees labels in $\mathcal{D}_s$, we call $\mathcal{D}_s$ the source domain and $\mathcal{D}_t$ the target domain. For the two encoders $f_c$ and $f_d$, we use Resnets [68] with 7 blocks trained with 100 iterations and a batch size of 128. Here $Z_c, Z_d \in \mathbb{R}^{256}$. We use $N' = 5,000$ samples in the max step for all methods.

**MNIST → MNIST-M**. Two digit datasets with the same classes but different background colors. Both datasets have 50,000 training samples. Table 5 shows the result, indicating that our method can more effectively remove the information about the domain. This is further confirmed by the T-SNE [69] plot in Figure 4, where one can hardly distinguish the samples of $Z_c$ from the two domains. This naturally leads to a higher target domain accuracy $\text{Acc}(\hat{Y}_t)$ than other methods.

**CIFAR10 → STL10**. Two datasets of natural images sharing 9 classes. There are 50,000 and 5,000 training samples in the two datasets respectively. Following existing works [70, 71, 47], we remove the non-overlapping classes from both datasets. Table 5 and Figure 4 show the result. Again, we see that our method can more effectively remove domain information from the learned representation.

## 6 Conclusion

This work proposes a new method for infomin learning without adversarial training. A major challenge is how to estimate mutual information accurately and efficiently, as MI is generally intractable. We sidestep this challenge by only testing and minimising dependence in a sliced space, which can be achieved analytically, and we showed this is sufficient for our goal. Experiments on algorithmic fairness, disentangled representation learning and domain adaptation verify our method's efficacy.

Through our controlled experiments, we also verify that adversarial approaches indeed may not produce infomin representation reliably – an observation consistent with recent studies. This suggests that existing adversarial approaches may not converge to good solutions, or may need more time for convergence, with more gradient steps needed to train the adversary fully. The result also hints at the potential of diverse randomisation methods as an alternative to adversarial training in some cases.

While we believe our method can be used in many applications for societal benefit (e.g. for promoting fairness), since it is a general technique, one must always be careful to prevent societal harms.

## Acknowledgement

AW acknowledges support from a Turing AI Fellowship under grant EP/V025279/1, The Alan Turing Institute, and the Leverhulme Trust via CFI. YC acknowledges funding from Cambridge Trust.

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
