# Supplementary Materials

## A Theoretical proofs

### A.1 Proof of Theorem 1

**Theorem 1.** Let $Z \in \mathbb{R}^D$ and $T \in \mathbb{R}^d$ be two random variables that have moments. $Z$ and $T$ are statistically independent if and only if $SI(Z;T) = 0$, where $SI(Z;T)$ is defined as follows[1]

$$SI(Z;T) = \sup_{h,g,\theta,\phi} \rho(h(\theta^\top Z), g(\phi^\top T)),$$

where $\rho$ is the Pearson correlation, $h, g : \mathbb{R} \to \mathbb{R}$ are Borel-measurable non-constant functions, and $\theta \in \mathbb{S}^{D-1}, \phi \in \mathbb{S}^{d-1}$ are vectors on the surfaces of $D$-dimensional and $d$-dimensional hyperspheres.

*Proof.* We first prove the direction $Z \perp T \Rightarrow SI(Z;T) = 0$, which is equivalent to prove $I(Z;T) = 0 \Rightarrow SI(Z;T) = 0$. By the data processing inequality, we know that $I(Z;T) = 0 \Rightarrow I(h(\theta^\top Z), g(\phi^\top T)) = 0, \forall h, g, \theta, \phi$. This immediately yields $\rho(h(\theta^\top Z), g(\phi^\top T)) = 0$.

We now consider the direction $SI(Z;T) = 0 \Rightarrow Z \perp T$. Since $\sup_{h,g,\theta,\phi} \rho(h(\theta^\top Z), g(\phi^\top T)) = 0$ for $\theta \in \mathbb{S}^{D-1}, \phi \in \mathbb{S}^{d-1}$, we know that $\forall h, g, \theta \in \mathbb{S}^{D-1}, \phi \in \mathbb{S}^{d-1}, \rho(h(\theta^\top Z), g(\phi^\top T)) = 0$. Now consider the moment generating functions. $Z \perp T$ if and only if $\forall t_1 \in \mathbb{R}^D, t_2 \in \mathbb{R}^d$

$$\mathbb{E}[e^{t^\top V}] = \mathbb{E}[e^{t_1^\top Z}]\mathbb{E}[e^{t_2^\top T}],$$

$$V_{1:D} = Z, \quad V_{D+1:D+d} = T, \quad t_{1:D} = t_1, \quad t_{D+1:D+d} = t_2$$

Assuming that now $\mathbb{E}[e^{t^\top V}] \neq \mathbb{E}[e^{t_1^\top Z}]\mathbb{E}[e^{t_2^\top T}]$ for some $t_1, t_2$, so that $\mathbb{E}[e^{t^\top V}] - \mathbb{E}[e^{t_1^\top Z}]\mathbb{E}[e^{t_2^\top T}] = \lambda \neq 0$. Then by setting $h, g, \theta, \phi$ as follows

$$h(\theta^\top Z) = \lambda e^{Z^\top \theta \cdot \|t_1\|}, \theta = \frac{t_1}{\|t_1\|}, \qquad g(\phi^\top T) = e^{T^\top \phi \cdot \|t_2\|}, \phi = \frac{t_2}{\|t_2\|}$$

we will have

$$\mathbb{E}[h(\theta^\top Z)g(\phi^\top T)] - \mathbb{E}[h(\theta^\top Z)]\mathbb{E}[g(\phi^\top T)] = \lambda^2 > 0$$

This suggests that $\rho(h(\theta^\top Z), g(\phi^\top T)) > 0$ for the chosen $h, g, \theta, \phi$ (since $h, g$ are not constant functions). However, this result contradicts with the condition $\forall h, g, \theta \in \mathbb{S}^{D-1}, \phi \in \mathbb{S}^{d-1}, \rho(h(\theta^\top Z), g(\phi^\top T)) = 0$. Therefore we must have $\mathbb{E}[e^{t^\top V}] = \mathbb{E}[e^{t_1^\top Z}]\mathbb{E}[e^{t_2^\top T}] \forall t_1 \in \mathbb{R}^D, t_2 \in \mathbb{R}^d$, meaning that $Z \perp T$. □

### A.2 Proof of Theorem 2

**Theorem 2.** Provided that each $h_i, g_j$ in (5) are $K$-order polynomials, given the sampled slices, we have $\hat{SI}(Z;T) \leq \epsilon \Rightarrow \sup_{i,j} \sup_{h_i,g_j} \rho(h_i(\theta_i^\top Z), g_j(\phi_j^\top T)) \leq \epsilon$.

*Proof.* We prove the contrapositive, i.e. rather than show $LHS \implies RHS$, we show that $\neg RHS \implies \neg LHS$.

Since each $h_i, g_j$ are $K$-order polynomials, we can rewrite $\sup_{h_i,g_j} \rho(h_i(\theta_i^\top Z), g_j(\phi_j^\top T))$ as

$$\sup_{w_i,v_j} \rho(w_i^\top Z_i', v_j^\top T_j')$$

---

[1] According to our definition $SI$ is always non-negative. This is because for any $h, g$ that satisfy $\rho(h, g) \leq 0$, we can always flip the sign of $\rho(h, g)$ by replacing $h$ by $-h$ or $g$ by $-g$, so that the value of $\rho(h, g)$ is higher.

whereas the expression of $\hat{SI}(Z;T)$ is

$$\sup_{w,v} \rho(w^\top Z', v^\top T')$$

where $Z'$ and $T'$ are fixed values (w.r.t $w$ and $v$) defined as follows:

$$Z'_i = [\sigma(\theta_i^\top Z)^k]_{k=1}^K, \qquad T'_j = [\sigma(\phi_j^\top T)^k]_{k=1}^K,$$

with $\sigma(\cdot)$ defined as in the main text l.103. Now assume that $\sup_{w_i, v_j} \rho(w_i^\top Z'_i, v_j^\top T'_j) > \epsilon$ for some $i, j$. Then by setting those elements in $w, v$ unrelated to $Z'_i, T'_j$ to zero, and those related to $Z'_i, T'_j$ exactly the same as $w_i, v_j$, we know that $\sup_{w,v} \rho(w^\top Z', v^\top T') > \epsilon$. This contradicts with $\sup_{i,j} \sup_{h_i, g_j} \rho(h_i(Z'_i), g_j(T'_j)) \le \epsilon$. $\qquad\square$

## A.3 Proof of the claim about the accuracy of polynomial approximation

**Proposition 3.** Consider approximating the supreme functions $h_i, g_j : \mathbb{R} \to \mathbb{R}$ in $\rho^*_{ij} = \rho(h_i(\theta_i^\top Z), g_j(\phi_j^\top T))$ by $K$-order polynomials, i.e.

$$h_i(a) \approx \hat{h}_i(a) = \sum_{k=0}^{K} w_{ik}\sigma(a)^k, \qquad g_j(a) \approx \hat{g}_j(a) = \sum_{k=0}^{K} v_{jk}\sigma(a)^k.$$

And assuming that $h_i, g_j$ and $\hat{h}_i, \hat{g}_j$ are the maximisers of $\rho(h_i(\theta_i^\top Z), g_j(\phi_j^\top T))$ and $\rho(\hat{h}_i(\theta_i^\top Z), \hat{g}_j(\phi_j^\top T))$ respectively. Without loss of generality we assume $\mathbb{E}[h_i] = \mathbb{E}[\hat{h}_i] = \mathbb{E}[g_j] = \mathbb{E}[\hat{g}_j] = 0$ and $\mathbb{V}[h_i] = \mathbb{V}[\hat{h}_i] = \mathbb{V}[g_j] = \mathbb{V}[\hat{g}_j] = 1$.

If $\forall i, j, a, |\hat{h}_i(a) - h_i(a)| \le \epsilon$ and $|\hat{g}_j(a) - g_j(a)| \le \epsilon$, then we will have $|\rho^*_{ij} - \hat{\rho}^*_{ij}| = o(\epsilon)$ where $\hat{\rho}^*_{ij}$ is the $K$-order polynomial approximation to $\rho^*_{ij}$.

*Proof.* For simplicity we below drop the subscripts $i, j$. Notice that

$$|\rho(h, g) - \rho(\hat{h}, \hat{g})| = |\mathbb{E}[hg] - \mathbb{E}[\hat{h}\hat{g}]| = |\mathbb{E}[hg - \hat{h}\hat{g}]| \le \mathbb{E}[|hg - \hat{h}\hat{g}|]$$

where the second equality comes from LOTUS. Now assume that $f = \hat{f} + \epsilon_1, g = \hat{g} + \epsilon_2$. Then

$$|hg - \hat{h}\hat{g}| = |-\epsilon_2 f - \epsilon_1 g + \epsilon_1\epsilon_2| \le |\epsilon_2||f| + |\epsilon_1||g| + |\epsilon_1\epsilon_2| \le \epsilon|f| + \epsilon|g| + \epsilon^2$$

which yields

$$\mathbb{E}[|hg - \hat{h}\hat{g}|] \le \epsilon\mathbb{E}[|f|] + \epsilon\mathbb{E}[|g|] + \epsilon^2 \le 4\epsilon + \epsilon^2$$

The last step is due to $\mathbb{E}[|f|] \le 2$, which holds whenever $\mathbb{E}[f] = 0$ and $\mathbb{E}[f^2] = 1$, as shown below:

$$\mathbb{E}[|f|] = \int_{-\infty}^{\infty} p(f)|f|df = \int_{-\infty}^{-1} p(f)|f|df + \int_{-1}^{1} p(f)|f|df + \int_{1}^{\infty} p(f)|f|df \le \mathbb{E}[f^2] + 1 = 2$$

Therefore $\mathbb{E}[|hg - \hat{h}\hat{g}|] = o(\epsilon)$ as $\epsilon \to 0$. This yields $|\rho(h, g) - \rho(\hat{h}, \hat{g})| = o(\epsilon)$ as $\epsilon \to 0$. $\qquad\square$

## A.4 Proof of the claim that a neural attribute predictor maximises a lower bound of MI

**Proposition 4.** Let $T \in \mathbb{N}$ be a discrete random variable. Training a predictor $\hat{T} = h(Z)$ by optimising the logistic regression loss: $J(h) = \mathbb{E}_{p(Z,T)}\left[\log \frac{e^{h_T(Z)}}{\sum_{k \neq T} e^{h_k(Z)}}\right]$ w.r.t $h$ is equivalent to maximise a lower bound of $I(Z;T)$ up to a constant $C$, i.e.

$$J(h) \le I(Z;T) - C, \forall h$$

*Proof.* Let $q_h(T|Z) = \frac{e^{h_T(Z)}}{\sum_{k \neq T} e^{h_k(Z)}}$. The loss above can then be rewritten as

$$J(h) = \mathbb{E}_{p(Z,T)}\left[\log q_h(T|Z)\right] = \int p(Z)p(T|Z)\log q_h(T|Z)dTdZ$$

On the other hand, we have

$$I(Z;T) = \int p(Z,T)\log\frac{p(T|Z)}{p(T)}dZdT = \int p(Z)\int p(T|Z)\log p(T|Z)dZdT + C$$

where $C$ is a constant unrelated to $h$. Subtracting the two formulas above, ignoring the constant $C$, we see that $I(Z;T) - J(h) = \int p(Z)\text{KL}[p(T|Z)\|q_h(T|Z)]dZ \ge 0$. $\qquad\square$

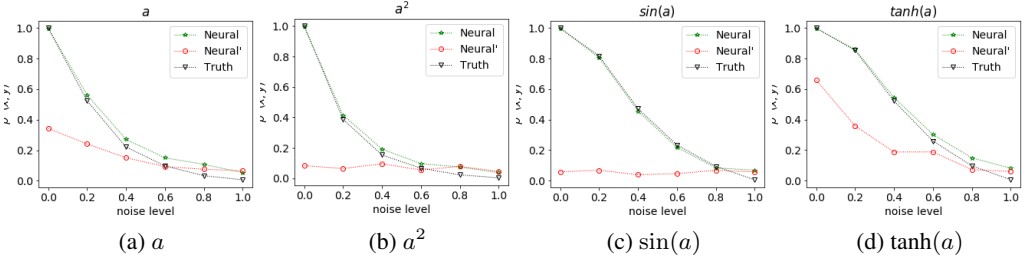

| (a) $a$ | (b) $a^2$ | (c) $\sin(a)$ | (d) $\tanh(a)$ |

Figure 1: The true Rényi correlation and its neural network estimate. x-axis is the noise level $\alpha$ (which controls the dependence between $X$ and $Y$) and the y-axis is the true/estimated values of $\rho^*(X, Y)$. Neural is the case when the neural network has been trained to converge and Neural' is the case when the network is trained using the same time as our method (i.e. it is not yet converged).

# B  Experiment details

We suggest readers to check the newest code on our github repo: `github.com/cyz-ai/infomin`.

## B.1  Details for neural networks

**Optimiser**. All neural networks are trained by Adam with its default settings and a learning rate $\eta = 0.001$. For adversarial approaches, we have also tried a learning rate of $0.01$ when training the adversary during minmax learning. The motivation for using a larger learning rate in the max step is that a larger $\eta$ can compensate for the (presumably) insufficient training time allocated. However, we discover that an overly large learning rate in the max step often hurts the quality of the trained adversary, therefore we recommend to use $\eta = 0.001$ for the both the min step and the max step.

**Early stopping**. Early stopping is an useful technique for avoiding overfitting, however it needs to be carefully considered when applied to adversarial methods. This is because in early stage of learning, the adversary $t$ in adversarial methods is often *not* trained thoroughly. In such cases, the estimate to validation error is often *underestimated*. If will use this inaccurate validation error to guide early stopping, we will prefer early iterations as the mutual information $I_t(Z; T)$ estimated by $t$ is low in these iterations. In fact, we find that disabling early stopping in adversarial methods often yields better performance. Due to these reasons we do not apply early stopping in adversarial methods. Our method does not has such issue, so in principle we can freely apply early stopping.

**Architectures**. Except stated otherwise, leaky ReLU is used as the nonlinearity in the hidden layer.

- *Networks used in the fairness tasks.* For the encoder $Z = f(X)$, we use a architecture of $[H, 10H, 10H, 80]$ where $H$ is the dimensionality of $X$. For the predictor $Y = r(Z)$, we use a architecture of $[80, 50, 50, |Y|]$. For the output layer of the encoder, we apply a non-linearity $\sin(\cdot)$ operation to map $Z$ to $[-1, 1]$. Other functions such as $\tanh(\cdot)$ can also be used and work similarly but empirically we found that $\sin(\cdot)$ works slightly better.

- *Network for computing Neural Rényi .* For the networks used to calculate Rényi correlation we use networks with 1 hidden layers with 200 hidden units;

- *Network for computing Neural TC.* For the network used to calculate total correlation we use an architecture of $[D + K, 100, 100, 1]$ where $D$ and $K$ is the dimensionality of $Z$ and $T$ respectively.

- *Feature adaptation layer in autoencoder.* We use a $[D + K, 200, 200, D + K]$ fully connected architecture for this network where $D$ and $K$ are the dimensionalities of $Z$ and $T$ respectively.

## B.2  Additional results and discussions

**On neural approximation to Rényi correlation**. We here investigate the reliability of using neural network to calculate Rényi correlation: $\rho^*(Z, T) = \sup_{h,g} \rho(h(Z), g(T))$ i.e. we approximate the supreme functions $h, g$ by two neural networks. In Figure 1, we show the true Rényi correlation for the independence test task considered in section 5.1 in the main text, along with their neural network approximation under different training conditions. Two conclusions can be made from the figure:

Table 1: Algorithmic fairness

| | USCensus | | | | | UCIAdult | | | |
|---|---|---|---|---|---|---|---|---|---|
| $S$ | 5 | 20 | 100 | 200 | $S$ | 5 | 20 | 100 | 200 |
| $\rho^*(Z;Y)$ | $0.95 \pm 0.00$ | $0.95 \pm 0.00$ | $0.95 \pm 0.00$ | $0.95 \pm 0.00$ | $\rho^*(Z;Y)$ | $0.98 \pm 0.01$ | $0.98 \pm 0.01$ | $0.98 \pm 0.01$ | $0.98 \pm 0.01$ |
| $\rho^*(Z;T)$ | $0.07 \pm 0.02$ | $0.06 \pm 0.02$ | $0.05 \pm 0.02$ | $0.07 \pm 0.02$ | $\rho^*(Z;T)$ | $0.21 \pm 0.06$ | $0.19 \pm 0.04$ | $0.15 \pm 0.03$ | $0.08 \pm 0.02$ |

Table 2: Disentangled representation learning

| | DSprites | | | | | CMU-PIE | | | |
|---|---|---|---|---|---|---|---|---|---|
| $S$ | 10 | 50 | 100 | 200 | $S$ | 10 | 50 | 100 | 200 |
| MSE | $0.46 \pm 0.04$ | $0.48 \pm 0.06$ | $0.55 \pm 0.02$ | $0.50 \pm 0.01$ | MSE | $1.92 \pm 0.11$ | $2.00 \pm 0.10$ | $2.05 \pm 0.09$ | $2.15 \pm 0.07$ |
| $\rho^*(Z;T)$ | $0.51 \pm 0.05$ | $0.26 \pm 0.04$ | $0.13 \pm 0.02$ | $0.08 \pm 0.02$ | $\rho^*(Z;T)$ | $0.30 \pm 0.03$ | $0.08 \pm 0.02$ | $0.05 \pm 0.01$ | $0.06 \pm 0.01$ |

Table 3: Ablation study: how the number of slices affects the performance of the proposed method.

- When trained thoroughly, the neural network approximation to Rényi correlation is fairly accurate, though it slightly overestimates $\rho^*(X, Y)$ in weak dependence scenarios (i.e. when $\alpha \geq 0.80$);

- When the network is not trained sufficiently (denoted as Neural' in the figure), the estimate bias to the truth is large and the estimate values are indistinguishable for different dependence level $\alpha$. This indeed reveals one typical failure mode of adversarial methods (insufficient training time).

From the figure, we also see that when $\alpha \geq 0.80$ the mutual information $I(X;Y)$ between $X$ and $Y$ is indeed very small, as can be seen from the low $\rho^*(X, Y)$ values in such cases which are $\leq 0.10$;

**Discussion on further baselines**. We here discuss how the considered baseline (in particular, Neural TC) is related to other baselines for infomin learning such as DANN/LAFTR. In DANN/LAFTR, a classifier is trained to distinguish whether a representation $Z$ belongs to the class $T = 0$ or $T = 1$, resulting in the following minmax objective for infomin learning:

$$\min_f \max_t \mathcal{L}(Z, Y) - \beta \cdot \mathbb{E}_{p(Z,T)}[\text{CE}_t(t(Z); T)]$$

where CE is the cross-entropy loss and $t(\cdot)$ is a classifier which classifies whether $T = 0$ or $T = 1$. The optimal classifier $t^*$ above satisfies (here $t_k(Z)$ denotes the softmax probability for class $k$)

$$t_k^*(Z) = p(T = k|Z)$$

This is indeed very related to the Neural TC baseline considered in our experiment. In neural TC, a classifier $t$ is trained to distinguish samples $(Z, T)$ from $p(Z, T)$ v.s. $p(Z)p(T)$:

$$TC(Z;T) = \mathbb{E}_{p(Z,T)}[CE_t(t(Z,T); 0)] + \mathbb{E}_{p(Z)p(T)}[CE_t(t(Z,T), 1)]$$

It can be shown that upon convergence, the optimal classifier $t^*$ satisfies:

$$t^*(Z, T) = \log \frac{p(Z, T)}{p(Z)p(T)} = \log p(T|Z) + C.$$

where $C = -\log p(T)$ is a constant unrelated to $Z$ and needs not to learn during network training.

By comparing the optimal $t^*$ in Neural TC and that in DANN/LAFTR, one can see that they are indeed equivalent to each other: both neural networks learns to estimate the conditional distribution $p(T|Z)$ either explicitly (DANN/LAFTR) or implicitly (Neural TC), despite the differences in loss functions and neural architectures. In addition, Neural TC is applicable to both discrete or continuous cases, whereas DANN/LAFTR can only be applied to the case where $T$ is discrete. Due to these reasons, we mainly compare with Neural TC rather than DANN/LAFTR in our experiments.

**Ablation study on the number of slices**. We investigate how the number of slices namely $S$ affect the performance of the proposed slice-based infomin learning method on two tasks. Table 1 and 2 show the results. From the tables, we see that in general, more slices $S$ leads to better information removal (as quantified by $\rho^*(Z;T)$). This is because with more slice we can more effectively detect dependence. We also see that $S = 200$ is generally sufficient to reach a low level of $\rho^*(Z;T)$ in various tasks. We therefore adopt $S = 200$ as the default setting and keep it fixed in all experiments.

Table 4: Algorithmic fairness: US Census

| | Neural Rényi | | | | Neural TC | | |
|---|---|---|---|---|---|---|---|
| $L_2$ | $\rho^*(Z,Y)$ | $\rho^*(Z,T)$ | time (sec./max step) | $L_2$ | $\rho^*(Z,Y)$ | $\rho^*(Z,T)$ | time (sec./max step) |
| 2 | $0.95 \pm 0.02$ | $0.23 \pm 0.10$ | 0.092 | 3 | $0.95 \pm 0.02$ | $0.27 \pm 0.03$ | 0.097 |
| 10 | $0.95 \pm 0.02$ | $0.19 \pm 0.06$ | 0.642 | 10 | $0.95 \pm 0.02$ | $0.21 \pm 0.02$ | 0.362 |
| 50 | $0.95 \pm 0.01$ | $0.06 \pm 0.02$ | 2.456 | 20 | $0.95 \pm 0.01$ | $0.08 \pm 0.02$ | 2.146 |
| Slice | $0.95 \pm 0.01$ | $0.07 \pm 0.02$ | 0.102 | Slice | $0.95 \pm 0.00$ | $0.07 \pm 0.02$ | 0.102 |

Table 5: Algorithmic fairness: UCI Adult

| | Neural Rényi | | | | Neural TC | | |
|---|---|---|---|---|---|---|---|
| $L_2$ | $\rho^*(Z,Y)$ | $\rho^*(Z,T)$ | time (sec./max step) | $L_1$ | $\rho^*(Z,Y)$ | $\rho^*(Z,T)$ | time (sec./max step) |
| 4 | $0.98 \pm 0.01$ | $0.17 \pm 0.08$ | 0.107 | 2 | $0.98 \pm 0.02$ | $0.36 \pm 0.13$ | 0.131 |
| 10 | $0.97 \pm 0.01$ | $0.12 \pm 0.06$ | 0.194 | 10 | $0.98 \pm 0.01$ | $0.20 \pm 0.03$ | 0.854 |
| 50 | $0.97 \pm 0.01$ | $0.06 \pm 0.02$ | 1.242 | 20 | $0.98 \pm 0.01$ | $0.07 \pm 0.03$ | 2.032 |
| Slice | $0.98 \pm 0.01$ | $0.08 \pm 0.03$ | 0.112 | Slice | $0.98 \pm 0.01$ | $0.08 \pm 0.03$ | 0.112 |

Table 6: Disentangled representation learning: Dsprite

| | Neural Rényi | | | | Neural TC | | |
|---|---|---|---|---|---|---|---|
| $L_2$ | MSE | $\rho^*(Z,T)$ | time (sec./max step) | $L_1$ | MSE | $\rho^*(Z,T)$ | time (sec./max step) |
| 6 | $0.61 \pm 0.04$ | $0.48 \pm 0.05$ | 0.791 | 8 | $0.49 \pm 0.03$ | $0.34 \pm 0.06$ | 0.812 |
| 50 | $0.87 \pm 0.05$ | $0.12 \pm 0.04$ | 1.903 | 50 | $0.83 \pm 0.01$ | $0.31 \pm 0.03$ | 2.013 |
| Slice | $0.50 \pm 0.01$ | $0.08 \pm 0.02$ | 0.602 | Slice | $0.50 \pm 0.01$ | $0.08 \pm 0.02$ | 0.602 |

Table 7: Disentangled representation learning: CMU-PIE

| | Neural Rényi | | | | Neural TC | | |
|---|---|---|---|---|---|---|---|
| $L_2$ | MSE | $\rho^*(Z,T)$ | time (sec./max step) | $L_1$ | MSE | $\rho^*(Z,T)$ | time (sec./max step) |
| 6 | $2.46 \pm 0.06$ | $0.36 \pm 0.04$ | 0.750 | 8 | $1.99 \pm 0.12$ | $0.39 \pm 0.06$ | 0.841 |
| 50 | $2.51 \pm 0.08$ | $0.26 \pm 0.06$ | 5.67 | 50 | $2.45 \pm 0.09$ | $0.36 \pm 0.05$ | 6.59 |
| Slice | $2.15 \pm 0.07$ | $0.07 \pm 0.01$ | 0.581 | Slice | $2.15 \pm 0.07$ | $0.07 \pm 0.01$ | 0.581 |

**Ablation study on the number of adversarial steps**. We here investigate how the number of gradient steps in inner loop optimisation (i.e. the hyperparameter $L_2$ in eq.2 and Algorithm 1) affect the performance of adversarial methods. Tables 4-7 show the results. The utility-fairness balancing factor $\beta$ for Neural Rényi and Neural TC are tuned in the way mentioned in the main text. The learning rate of the adversarial steps is the same as that of the outer loop and is set to be $\eta = 0.001$.

Overall, we see that the proposed slice method is more scalable than the two adversarial training methods. To achieve the same level of fairness/disentanglement, adversarial methods typically require a much longer time. For fairness tasks, we see that adversarial training methods can catch up with our method when sufficient time budget is given. For disentanglement tasks, this is not true, as by increasing the time budget adversarial methods can still not outperform our method. We conjecture this is due to optimisation difficulties in minmax problem. The results identify two typical failure modes of adversarial methods: (a) insufficient training time and (b) optimisation difficulties.

**Details for tuning the v-CLUB method**. We here share some experience for tuning the v-CLUB method, which is a strong baseline but needs careful hyperparameter tuning. We discover that a larger batch size (e.g. $\geq 256$), a smaller learning rate (e.g. $\eta = 0.0001$) and a smaller balancing factor (e.g. $\beta \leq 0.01$) are generally more preferable in v-CLUB. We conjecture that this is because when the batch size is small, the estimation variance of mutual information $\mathbb{V}[\hat{I}(Z;T)]$ in v-CLUB is high (which is a known issue of most neural mutual information estimators), resulting in unstable optimisation. Such optimisation instability will further be strengthen if $\eta$ and $\beta$ are not small enough.