# OpenReview forum: "Scalable Infomin Learning"
_NeurIPS.cc/2022/Conference — NeurIPS 2022 Accept_

### Official Review · Reviewer_hqSG · 2022-07-09

**Rating:** 7
**Confidence:** 3
**Soundness:** 3 good
**Presentation:** 3 good
**Contribution:** 3 good

**Summary:**

The authors propose a method for infomin learning based on information theory tools from previous work, that estimates the Mutual Information (MI) in the “sliced space”, which intuitively is a facet of MI.

**Questions:**

- On the disentanglement experiment more recent work could be considered as baseline, for example LieGroupVAE (https://arxiv.org/pdf/2106.03375.pdf) ?
- In the disentanglement experiment qualitative Figure2-3. Should latent traversals also be explored, to understand if the representation is disentangled?
- In the fairness experiment ρ∗(Z, T ) seems to get non-comparable results for the baselines. Should other better performing methods be considered as baselines?


**Limitations:**

One of the claims is the scalability, however a dedicated experiment demonstrating how the model scales compared to previous baselines is missing.


**Strengths And Weaknesses:**

Strengths
- The method seems explained well in terms of theoretical justifications
- The experiments seem to be provided with sufficient details

Weaknesses
- In the disentanglement experiment other metrics could be used to make sure that models are learning disentangled representations, e.g. MIG, factorVAE score, etc

---

> ### Author Response · Authors · 2022-08-02
> **Thanks for your comments; clarifications on evaluation metrics/baselines; new experiment added**
>
>
> Thank you very much for your valuable feedback. Below we try to address your concerns and questions.
>
> To your comment regarding the weakness of the work:
>
> - We agree that adopting metrics in existing literature like MIG or Factor-VAE score will be very helpful but they might not be so suitable here. This is because these metrics are mainly designed to assess how each dimension $Z_d$ (a scalar) in the representation $Z$ is disentangled from each other, whereas our scenario here is to assess the disentanglement between two vectors ($Z$ and $T$). In fact, if one adapts MIG to our scenario, he/she will see the that MIG is indeed equivalent to the metric we used (we can elaborate more on this). On the other hand, we have also tried to adapt Factor-VAE score to our case. However, this metric needs to find the most invariant dimension in controlled generation, but how this invariance is defined for vectors remains unclear. Nonetheless we will continue to try other ways for adapting these metrics to our case.
>
> To your questions:
>
> - Thanks for pointing us to LieGroup VAE, which broadens our view. This kind of group theory-based method for achieving disentanglement is definitely worth mentioning, and we have now included a discussion about it (and other related works) in the revised paper (see p5 in the main text). However, as our focus here is to compare different information-theoretic approaches (as represented by the current baselines), we regret that we will not directly compare with LieGroup VAE (or other group theory-based methods) this time, but rather leave it to future work, where the comparison between information theory-based and group theory-based approaches for disentangelment will be explored.
>
> - Suggestions on latent traversal: the label swapping experiment presented in Figure 2 and Figure 3 is indeed a kind of latent traversal. More specifically, if we travel horizontally among the images in the same row, then it is the same as "fix $T$ and change $Z$" (where the sampling distribution $p(Z)$ is implicitly defined by the training set). On the other hand, if we compare the images before and after label swapping, then it is the same as "fix $Z$ and change $T$" (where the sampling distribution $p(T)$ is a uniform distribution). We have revised the captions in Figure 2 and Figure 3 accordingly to better inform the readers.
>
> - Baselines too weak: the seemingly poor performance of the considered baselines is indeed the result of our controlled experiment where the time budget of the baseline is set to be the same as our method. In fact, if we increase the time budget of these baselines (i.e. increase the number of gradient steps in inner loop optimisation) unlimitedly, the performance of these baselines can catch up with our method or even outperform it. See the new ablation study on p5 in the revised appendix. On the other hand, we do have included a new, strong baseline i.e. Variational Upper Bound (denoted as VB in the main text) in the revised paper. Please see the paper for more details.
>
> - Experiments on how the new method scales compared to previous baselines: in this work, we mainly define scalability as the time complexity of the algorithm required to reach a certain level of performance.  In adversarial methods, this is $O(L_1 L_2)$ whereas in our adversary-free method this is $O(L_1)$. Here $L_1, L_2$ are the number of gradient steps for outer and inner loop optimisation respectively. To verify that the proposed method is really more scalable, a new ablation study on how the performance of adversarial approaches change w.r.t $L_2$ has now been included; see "ablation study on the number of adversarial steps" on p5 of the updated appendix. From the new experiments, we see that in order to achieve the same level of performance as our method, adversarial methods typically require a large $L_2$ and hence much longer execution time than our method. We believe this demonstrates the scalability of our method.

---

> > ### Comment · Area_Chair_hzHT · 2022-08-07
> > **Reviewer hqSG: Could you comment on the author response?**
> >
> > *Reviewer hqSG*: Could you comment on the author response? In particular, do you agree that disentanglement metrics are not applicable to vector-valued variables? Does the new ablation study on p5 address (some of) your concerns regarding the strength of baselines and scalability?
> >
> > I would also appreciate if you could expand upon your overall opinion of this submission; you review in its current form is rather terse, which makes it difficult for the meta-reviewers to evaluate whether this submission should be strongly considered for acceptance.
> >
> > Thanks!

---

> > > ### Comment · Reviewer_hqSG · 2022-08-08
> > > **Updated score from 6 to 7**
> > >
> > > The author's answer and updated work address my concerns. The disentanglement evaluation setup based on vector rather than scalars seems sensible - and it seems that traditional metric could be adapted into this setup. The new ablation study and author's reply addressed and clarified my scalability question.

---

### Official Review · Reviewer_dNaL · 2022-07-10

**Rating:** 7
**Confidence:** 4
**Soundness:** 3 good
**Presentation:** 3 good
**Contribution:** 3 good

**Summary:**

The authors propose an adversary-free method for informin-based representation-learning based on the recently-proposed sliced mutual information estimator, noting the optimisation difficulties and costs associated with the adversarial approaches that have long been the go-to methods for tasks such as disentanglement and fair-representation learning. This is achieved by substituting the empirical approximation of sliced mutual  information (SI) with a closed-form solution that can be well-approximated with K-degree polynomials, leading to a problem that can be efficiently solved using canonical correlation analysis. The authors establish error bounds on said analytic approximation and show that the entailed functions can be solved in parallel for each slice without violating the objective.
Experiments are performed on a number of datasets spanning the interrelated tasks of independence  testing, disentanglement, and fair-representation learning, with Renyi correlation between the learned representation and Y (aiming to maximise) and T (aiming to minimise) as the evaluation metric. The proposed method, Slice, is shown to  outperform other infomin-learning approaches by a significant
margin while remaining computationally efficient.


**Questions:**

-

**Limitations:**

I am satisfied with the extent to which the the limitations and ethics of the work have been addressed.

**Strengths And Weaknesses:**

- The paper builds on recent research on mutual-information estimation to develop a method that is both adversarial free
and computationally-efficient, and how it might be practically applied to problems of disentanglement and fair-representation
learning -- to my knowledge, the analytic approximation, derived from canonical correlation analysis, is novel within the
given context, and seems to be both theoretically and practically sound.
- The motivation for the seeking out non-adversarial alternatives to infomin learning is clearly established and the description of the method is generally well-structured and easy-to-follow, though I feel there is some room for improvement in the choice of notation (such as in the decision to use D and d for the dimensionality of Z and T, respectively).
- The analysis of the results is satisfactory; the conclusion is serviceable but feels rushed and I'm not convinced there's sufficient evidence given in the paper to 'verify' the failure modes of adversarial methods.
- The method is benchmarked against a good range of relevant baselines and using an appropriate suite of datasets
(though it would be nice to see variation in the domain of the fairness datasets) and results are aggregated over multiple
replicates in all cases. The procedures associated with each set of experiments are outlined in sufficient detail.
- While the existing baselines are solid, for the fair-representation learning experiments it would be nice to see a simpler
adversarial cross-entropy-based baseline a la DANN/LAFTR, given its prominence in the literature, despite its inability
to capture higher order dependencies like its Renyi counterpart.
- Given the statements made about the convergence of the adversarial methods, I feel it would also be of interest to include ablations in which the number of adversarial steps is increased enough to determine whether this is a relevant factor, or whether the problems with these methods truly are of a more fundamental nature.
- The method, Slice, seems to perform impressively in terms of both expectation and variance. While Slice does incur decent computational overhead compared the simpler baselines, such as 'Pearson', it is nonetheless shown to be significantly more efficient than its adversarial competitors which suffer additional problems due to their parametric/minimax nature. The results included in the main paper, together with those included in the Appendices, support these conclusions convincingly enough, with the experimental setups used to produce them largely consistent with that of past work.
- While the authors do discuss the relationship between Renyi correlation and DP, and good rationale for adopting solely the former, it would nonetheless be good to additionally include the the latter for the evaluation of the method in the context of representation learning, given the conventions of the pre-existing literature on the problem.
- The images in Figures 2 and 3 comparing how Slice and an adversarial method (which one?) cope with label-swapping are rather small and there's no indication as to what the label corresponds to within the figures themselves or their captions. Those things aside, I think it would be helpful if the differences between the methods were highlighted as it's not immediately obvious where exactly one should be looking.

---

> ### Author Response · Authors · 2022-08-02
> **Many thanks for appreciating our work; it has been improved according to your suggestions**
>
> We are thankful for your positive comments as well as the suggestions. They are really insightful and help a lot in improving the work.  Following your suggestions we have made several improvements. Please see below.
>
> - As you kindly suggested, we have now included a new ablation study on how the number of adversarial steps will affect the performance of adversarial methods. The results for the fairness tasks are presented on p5 in the updated appendix (the same experiment for disentangled representation learning is still running and is really time-consuming). Two conclusions can be made: (a) given sufficient time budget, adversarial methods can indeed catch up with or even outperform our method, so their poor performance here may be just due to insufficient training time rather than other factors such as their neural/minmax nature; (b) however, to reach the same level of fairness, the required time in adversarial methods are much longer than our method, typically several times or even an order longer. Note that we are conservative in generalising these conclusions above to other tasks/datasets. More comprehensive reports will be given after collecting the results of disentanglement tasks.
>
> - Together with the analysis in the appendix, the ablation study above may also serve as a verification of one typical failure mode (non-sufficient training time) of adversarial training methods. We will continue to verify other failure modes (e.g. optimisation difficulty).
>
> - We have also investigated in depth DANN/LAFTR, two important baselines you mentioned. Although they have different motivation and loss functions, we found that these cross-entropy-based methods have many similarities to Neural TC, a baseline we already considered. More specifically, the neural network in both methods estimates the conditional density $p(T|Z)$ either explicitly (DANN/LAFTR) or implicitly (Neural TC). See the analysis “discussion on other potential baseline” in p5 of the updated appendix. In this sense we feel it is unnecessary to compare with DANN/LAFTR but just mention them. Another reason we choose not to compare with DANN/LAFTR is that they can only be applied to the case where $T$ is discrete, whereas the tasks considered here cover both continuous and discrete cases. Neural TC, on the contrary, does not have such limitations and can be applied in all cases. We have added the above discussion in the revised manuscript.
>
> - Other suggestions such as improving notation and the appearance of figures, and including commonly-used evaluation metrics will also be addressed very soon.

---

> > ### Comment · Area_Chair_hzHT · 2022-08-07
> > **Reviewer dNaL: Could you comment on the author response?**
> >
> > *Reviewer dNaL*: The authors have followed up to determine how the performance of adversarial methods depends on the number of adversarial steps. Could let them know what you think of these additional experiments? I would also appreciate hearing whether you agree that the Neural TC baseline is sufficiently similar to DANN/LAFTR. Thanks!

---

### Official Review · Reviewer_d477 · 2022-07-15

**Rating:** 6
**Confidence:** 3
**Soundness:** 3 good
**Presentation:** 3 good
**Contribution:** 3 good

**Summary:**

In this paper, the authors leverage sliced mutual information and propose a tractable infomin learning algorithm. Compared to the existing methods, the proposed algorithm requires neither adversarial training nor a neural-based mutual information estimator. Instead, the authors replace the mutual information term with a sliced version in the infomin objective, and derive an analytic approximation to sliced mutual information. Moreover, solving hi and gj in the sliced mutual information together yields an upper bound of the analytic solution, and thus more efficient in practice. The authors compare the proposed method with 4 different baselines, including the non-parametric method and adversarial training method. They evaluate them on three tasks: in the independence test, they show that sliced mutual information is less effective than the adversarial training methods, but they require longer training time; in the fairness experiment, under a similar level of utility, the proposed method has significantly better fairness than the baselines; in the disentangled representation learning experiment, they show a similar observation to the fairness experiment that the proposed method achieves much better disentanglement without sacrificing the utility a lot.

**Questions:**

Please answer the questions in the weaknesses section.

Both your method and MI upper bound [1] utilize neural networks for MI estimation, could the authors elaborate more on the advantages of your method?

I am willing to increase my scores if my questions and concerns are resolved.


**Limitations:**

See the weakness section.

**Strengths And Weaknesses:**

**Strengths**:
- The paper is overall well written. The theoretical analysis is sound and well motivated.
- The idea of using proxy metrics instead of directly working on the intractable MI is interesting
- The experimental setup is sound with different baselines

**Weaknesses**:
- I think the paper misses one important and related paper [1], and thus lacks further discussion of one important implementation of infomin optimization that leverages neural-based methods to give a trackable estimation of the upper bound of mutual information.   This should serve as a strong baseline and compare with the proposed method, non-parametric methods, and adversarial training methods as well. I am expecting to see more quantitative analysis on the comparisons.
- I am also looking forward to seeing more larger-scale experiments, as the paper is claimed to be “scalable”. For example, the CLUB MI estimator [1] is known to be applied to large language models like BERT to improve the adversarial robustness [2].
- The paper misses some important implementation details: (1) a brief introduction and citation to the datasets Dsprite and CMU-pie are missing. (2) how the networks h,g are trained needs more details, e,g, the training objective. (3) I am also looking forward to seeing a detailed pseudo algorithm in terms of the details mini-batch learning to improve the reproducibility.
- I am also expecting to see how is the sliced MI related to the real MI. It might be better to give some toy examples where MI can be calculated and see how close the methods are.

[1] Cheng, Pengyu, Weituo Hao, Shuyang Dai, Jiachang Liu, Zhe Gan and Lawrence Carin. “CLUB: A Contrastive Log-ratio Upper Bound of Mutual Information.” ICML (2020).

[2] Wang, Boxin, Shuohang Wang, Yu Cheng, Zhe Gan, R. Jia, Bo Li and Jingjing Liu. “InfoBERT: Improving Robustness of Language Models from An Information Theoretic Perspective.” ICLR (2021)

---

> ### Author Response · Authors · 2022-08-01
> **New baseline included; our method is indeed not a new MI estimator**
>
> Many thanks for your detailed comments and the criticism. They identify some important weaknesses of the work which we hope to address here and in the revision. We would also like to clarify some points which may have been misunderstood due to our presentation.
>
> Clarifications:
>
> a) *SI is not an estimator to MI nor its bounds*, but a proxy of MI used in optimisation. While SI shares the same motivation as CLUB (to achieve infomin learning), and that they both have the property "SI=0/CLUB=0 -> MI=0", the principles behind them are quite different. More specifically, CLUB works by first estimating (an upper bound of) MI then minimising it. SI, on the contrary, never estimates MI (or its bound), but instead minimises statistical dependence in the sliced spaces. The difference here is “MI estimate in the original space” v.s. “independence test in the sliced space”. Importantly, working in the sliced space allows us to find an analytical expression for SI, which is not possible for CLUB (and other neural MI estimators). A demo for demonstrating how SI works as compared to CLUB has been included in the uploaded code.
>
>
> b) *SI is neural network-free*. While it is natural to model the functions $h, g: R \to R$ in SI by neural networks, they are modelled by K-degree polynomials (remark that $h, g$ are 1D functions so polynomials are generally powerful enough). For a particular slicing direction, the parameters (i.e. the coefficient) of the polynomials for that direction can be solved analytically by eigendecomposition, as explained by the texts from eq.5 to eq.6 on p3. No neural network is used here. That being said, it is also possible to model $h, g$ as neural networks, which are more powerful but do not have analytic solution any more (and in such case one have to learn them by SGD).
>
> Responding to the commented weakness:
>
> - We thank you very much for pointing us to CLUB, an important baseline that we should not have missed. We have now compared to CLUB. Please see the revised manuscript. Code for reproducing the results has also been updated (which is adapted from the author’s official repo). We find that CLUB does work very well in many cases, being a strong baseline to compare with (possibly due to its upper bounding nature). However, it is still less effective than our method under the time budget given. One possible explanation is that CLUB is still in essence an adversarial approach (where the conditional density $p$ needs to be estimated by some gradient steps first), and the tightness of the upper bound in CLUB depends on how good $p$ is. If the number of gradient steps used to learn $p$ is insufficient, or $p$ is not powerful enough, the resultant bound may not be tight and hence less satisfactory performance. Nonetheless, given sufficient training time and powerful enough $p$, it is completely possible that CLUB will outperform our method.
>
> - We also agree that it is beneficial to include large-scale experiments such as those considered in InfoBert, where the model size is large and the dimensionality of representation is high. However, besides these factors, scalability may also be evaluated by the time complexity of an algorithm. From this viewpoint, an adversarial training-free method is clearly more scalable than its adversarial counterpart as it scales the time complexity from $O(L^2)$ to $O(L)$ (here $L$ is the number of gradient steps). In fact, in a new ablation study, we find that to achieve the same level of performance, our adversary-free method typically requires much less time than adversarial methods; see p5 "ablation study on the number of adversarial steps" in the new appendix. So we believe our claim on scalability is still reasonable.
>
> - Thanks a lot for pointing out these issues. We have now correctly cited the two datasets as well as provided an algorithm block in the main text (see p4 in the main text). To ease comparison, we have also provided an algorithm block for conventional adversarial approaches. For the learning of the functions $h, g$, they are not neural networks and are solved analytically using objective (6), as already clarified in b) above.
>
>
> - As clarified in a), SI is just a proxy of MI used in optimisation rather than an estimate to it (they even do not work in the same space). It is neither the lower nor upper bound of MI. In this sense, it may make little sense to compare the values of SI and MI directly as done in the CLUB paper. Investigating the test power of SI as an independence test as done in the current experiment may be more sensible.
>
> Responding to your question:
>
> As clarified in b), there is no neural network used in our method: the functions $h, g$ in SI are approximated by polynomials whose parameters can be solved analytically. This analytical property, which removes the need of adversarial training, makes our method fundamentally different to other neural network-based approaches, and is the source of scalability of our method.

---

> > ### Comment · Area_Chair_hzHT · 2022-08-07
> > **Reviewer D477, have the authors addressed (some of) your concerns?**
> >
> > *Reviewer D477*: The authors have added a comparison to CLUB, as well as an additional ablation study to support the claim of scalability. Could you let the authors know to what extent your concerns have been addressed?

---

> > > ### Comment · Reviewer_d477 · 2022-08-08
> > > **Thank you for the response!**
> > >
> > > Thanks for the authors' response.
> > >
> > > I am glad to see that the presentation of the paper significantly improves, and the authors add a pseudo algo comparison with adversarial learning based approaches as well as the CLUB baseline. I am mostly satisfied with the answers. Thus I update the score from 5 to 6.

---

### Author Response · Authors · 2022-08-02
**Summary of the response**

We would like to thank all reviewers and the AC for their time and the insightful comments. The comments are highly valuable and help greatly to improve the quality of the work. Below is a summary of our response:

- We appreciate all reviewers for acknowledging the technical novelty and theoretical soundness of the method. The proposed method presents a third way (i.e. random independence test) for achieving infomin representation learning where the choices have long been either adversarial- or variational- methods. A new, detailed analysis on how the method differs to recent baseline suggested by the reviewers (reviewer d477) are also included in the rebuttal.

- We also thank all reviewers for pointing out the limitations of our work in empirical evaluation, most of which as we believe have now been addressed. The efforts include comparison with further baselines (requested by reviewer d477, dNaL and hqSG), new ablation studies regarding the scalability of the proposed method and the baselines (requested by reviewer dNaL and reviewer hqSG), clarifications on the evaluation metrics used, better reproducibility (code demo + better presentation) etc. Please see the one-to-one response below.

Any further comments/criticisms/suggestions are welcome.

-------------------------------------------------------------------------------

We sincerely thank the reviewers for their post-rebuttal feedback, as well as the AC who has been very responsible.

---

### Meta-Review · Area_Chair_hzHT · 2022-08-30

**Recommendation:** Accept
**Confidence:** Less certain

**Metareview:**

**Summary**: This paper develops an infomin-based representation method that based on the recently-proposed sliced mutual informaiton estimator. Unlike other methods, the proposed approach does not rely on an adversarial objective and provides tractable proxy-metric that eliminates the need for neural estimators of the mutual information. Experiments on independence tests, disentangled representation learning and algorithmic fairness aim to illustrate both improved utility and higher scalability.

**Strengths**: Reviewers we overall positively predisposed towards this paper. They noted that this is a well-written paper, with sound and well-motivated theoretical analysis [d477, hqSG]. The proposed method, which derives from canonical correlation analysis is novel and computationally efficient [d477]. Experiments are sound and satisfactory, with benchmarks that include a good range of datasets and baselines. Reviewer *dNaL* notes good results in terms of both expectation and variance, in addition to improved computational efficiency relative to adversarial methods.

**Weaknesses**: Reviewers also noted limitations. Reviewer *dn477* noted a missing reference to CLUB (Chen et al., ICML, 2020) which would be a strong neural baseline. Several reviewers found that scalability claims are not strongly supported and that larger-scale experiments might strengthen the paper in this context [d477, hqSG]. More generally, reviewers were concerned that the submission lacks certain important implementation details [d477, dNAL]. In terms of the experiments, reviewers had a number of suggestions, including a comparison between slice MI to the analytically calculated MI for some toy example [d477], a comparison simpler a simpler adversarial cross-entropy based methods for the fairness experiments (e.g. DANN/LAFTR) [dNaL], a comparison to LieGroupVAE [hqSG], and reporting of disentanglement metrics such as MIG and the FactorVAE score [hqSG].

**Reviewer Author Discussion**: While the authors were not able to carry out larger-scale experiments, they provided an ablation study to further support claims of scalability.  They also added discussion of CLUB, and clarified that DANN/LAFTR are similar to the Neural TC baselines, clarified that disentanglement scores cannot be computed for the vector-valued quantities that are under consideration for this paper. Reviewer *D477* raised their score 5->6, reviewer *hqSG* raised their score 6->7.

**Reviewer AC Discussion**: Reviewers unfortunately did not respond to the AC during the discussion phase. The AC takes this as a signal that reviewers do not object to acceptance, but also do not champion it.

**Overall Recommendation**: This submission is just about above the bar for an accept, though lack of a clear champion among reviewers somewhat limits confidence.

**Award:**

No

---

### Decision · Program_Chairs · 2022-09-14

Accept